# Computer Vision for Glass Waste: Technologies and Sensors

**DOI:** 10.3390/s25216634

**Published:** 2025-10-29

**Authors:** Eduardo Adán, Antonio Adán

**Affiliations:** Department of Electrical Engineering, Electronics, Automation and Communications, University of Castilla La Mancha, 13002 Ciudad Real, Spain; eduardo.adan1@alu.uclm.es

**Keywords:** computer vision, glass, recycling, waste

## Abstract

Several reviews have been published addressing the challenges of waste collection and recycling across various sectors, including municipal, industrial, construction, and agricultural domains. These studies often emphasize the role of existing technologies in addressing recycling-related issues. Among the diverse range of waste materials, glass remains a significant component, frequently grouped with other multi-class waste types (such as plastic, cardboard, and metal) for segregation and classification processes. The primary aim of this review is to examine the technologies specifically involved in the collection and separation stages of waste in which glass represents a major or exclusive fraction. The second objective is to present the main technologies and computer vision sensors currently used in managing glass waste. This study not only references laboratory developments or experiments on standard datasets, but also includes projects, patents, and real-world implementations that are already delivering measurable results. The review discusses the technological possibilities, gaps, and challenges faced in this specialized field of research.

## 1. Introduction: Glass Waste and Recycling

### 1.1. A Brief Introduction to Global Solid Waste

The continuous increase in global waste generation represents one of the most pressing environmental challenges of our time. Each year, approximately two billion metric tons of waste are produced worldwide, and this figure is expected to rise sharply in the coming decades, placing additional pressure on waste management systems [1]. Although recycling plays a crucial role in mitigating this impact, it is insufficient on its own. Broader strategies are required, including product designs that enhance circularity, improvements in collection systems, and, above all, the automation of material recovery through emerging technologies.

According to international agencies such as the World Bank [2] and the U.S. Environmental Protection Agency (EPA) [3], global waste can be broadly classified into three main categories: Municipal Solid Waste (MSW), Industrial Waste, and Construction and Demolition Waste (C&DW). Glass is commonly found in all of them, usually mixed with other materials depending on the specific waste stream.

MSW accounts for roughly 45–55% of the total waste generated globally and includes domestic, commercial, and institutional refuse—food scraps, paper, plastics, metals, and notably, glass. In the United States, about 12.25 million metric tons of glass waste were produced in 2018 (4.2% of MSW), with approximately 31.3% recycled [3]. Industrial waste, representing 20–25% of total waste, originates mainly from manufacturing, mining, and energy-related activities. Although comprehensive data on glass within Industrial Solid Waste (ISW) are limited, reports indicate that fine glass (<1 mm) can constitute about 13% of processed waste, while larger glass and ceramic fractions (>12 mm) may reach 46% [4].

Within the C&DW category—responsible for 20–30% of global waste—glass appears in smaller amounts together with materials such as concrete, metals, and wood. In the European Union, approximately 4% of total glass waste (around 1.5 million tons) originates from construction and demolition activities [5]. In contrast, electronic waste, though representing less than 3% of global waste volume, contains significant glass components: in India, it may include up to 0.30% glass and 19% lead glass from cathode ray tube (CRT) devices [6]. Finally, agricultural waste, estimated at 5–10%, mainly comprises organic matter and plastics, with no evidence suggesting any meaningful glass content.

### 1.2. Representative Figures on Glass Waste Generation and Recycling

Figure 1 provides information on the main sources or environments of glass waste. A brief description is given for each waste type, along with the commonly generated glass forms in each case. Current global glass production is approximately 130 million metric tons per year, according to previous studies in [7]. This production is composed of container glass (48%), flat glass (42%), tableware (5%), and other glass products (5%). It is estimated that the European Union produces around 40 million metric tons of glass products annually [8], while the United States reaches an annual production of 12.3 million metric tons [3].

As previously mentioned, glass accounts for approximately 4% to 8% of municipal solid waste generated globally, with variations depending on per capita consumption and each country’s waste management system. Individually, industrialized countries generate between 10 and 25 kg of glass waste per capita annually [3]. For example, in the European Union, over 11 million metric tons of glass are selectively collected each year [9].

Although glass is 100% recyclable and can be reprocessed indefinitely without quality loss, recycling rates vary significantly across regions. According to Ran et al. (2016) [10], the global glass recycling rate is around 50%. More recent studies (2025) [11] indicate that in the European Union, the average recycling rate for glass packaging is approximately 80%. Countries such as Germany (85.3%), France (83.7%), and Spain (75.9%) are notable examples of high recycling performance due to highly efficient selective collection systems (data provided by Eurostat [12]). In contrast, the United States exhibits a much lower recycling rate—around 34% in 2025 [11]—attributed to limited source separation and less efficient waste management systems [3]. In regions such as Latin America and Asia, glass recycling largely depends on informal labor, with estimated rates between 35% and 50%, although official data remain scarce [13]. Brazil, a representative country in Latin America, reported a 47% glass recycling rate in 2025 [14]. In Asia, Japan and South Korea reported the highest rates—71% and 86%, respectively, in 2025 [11]. Beyond diverting waste from landfills, glass recycling offers key environmental benefits, including energy savings of up to 30% compared to production using virgin raw materials, a significant reduction in CO_2_ emissions, and the conservation of natural resources such as sand and limestone.

Figure 2 presents a comparative chart illustrating the estimated glass recycling rates in several representative countries and continents. It clearly highlights the differences between countries such as Germany (85.3%) and the United States (34%), as well as the estimated ranges for countries of Latin America.

## 2. Objectives, Contributions, and Structure of the Paper

In the context of glass collection and recycling, computer vision systems represent a strategic tool to improve efficiency in waste management. Their application enables the automation and optimization of processes, as well as the classification and separation of recyclable materials, reducing human errors and increasing the recovery rate of valuable resources [15]. By utilizing image sensors and artificial intelligence algorithms, these technologies can identify and categorize various types of waste in real time with high levels of accuracy (the term glass accuracy is defined as the percentage of correctly identified glass items with respect to the total number of actual glass instances in the input stream [16]). Their implementation in recycling plants, production lines, and smart collection points not only contributes to more effective management but also reduces the volume of waste sent to landfills and incinerators. Furthermore, when integrated with other intelligent systems, computer vision can support predictive analytics platforms and traceability, facilitating data-driven decision-making.

Although several reviews on solid waste and recycling ([17,18,19]) have been published, generally covering all types of materials, there are hardly any review articles focusing specifically on glass. Only the article by Baek et al. ([11]) can be found, in which little attention is given to imaging acquisition technologies or computer vision systems for glass classification; likewise, it does not provide a clear taxonomy of optical sensors or specify performance metrics. The aim of this article is to review the state of the art in glass collection and recycling, highlighting the latest applied technologies, and subsequently focusing on computer vision–based approaches. Therefore, the novelty of this review stems from this gap. Furthermore, its relevance is reinforced by the fact that glass constitutes approximately 10% of the world’s waste material.

What this review adds is not only a taxonomy of imaging sensors organized by task and setting, highlighting the most effective computer vision technologies and devices currently used in glass collection and sorting processes, but also an analysis of their limitations, gaps, and shortcomings. The study of AI-based methods and algorithms grounded in computer vision is beyond the scope of this review [17].

The structure of the paper begins with an overview of solid waste, introducing glass waste as one of its specific categories. The following section explores glass waste in greater detail, providing relevant statistics together with a discussion of its main sources, thereby helping the reader to appreciate the importance of its management. Section 3 focuses on the methodological aspects of the review process, while Section 4 offers an overview of the available equipment and technologies. Section 5 presents the most recent technologies applied to glass waste, both in the collection phase and in the classification stage. Section 6, which constitutes the core of the article, offers a comprehensive review of computer vision technologies, highlighting their potential applications, advantages, and limitations. Section 7 includes a concise introduction to software technologies, which will be the focus of a forthcoming review. Finally, Section 8 discusses the challenges, recommendations, and future directions in the application of computer vision to improve glass waste treatment. Additionally, ways to promote the use of glass from institutional, technological, and citizen participation perspectives are proposed.

## 3. Methodological Aspects of the Review

The foundation of any bibliographic review relies heavily on the quality of the inputs. For this reason, an initial survey of the literature was conducted, covering topics related to glass waste and the technologies applied across the different stages of its treatment. Specifically, searches were carried out in the areas of waste acquisition or collection, classification and recognition, and glass recycling. These results were subsequently refined in order to establish more specific objectives, focusing on sensors and techniques associated with robotics and automation. This process led to the formulation of four search strings, as summarized in Table 1.

In this study, the databases Scopus, Web of Science, and Google Scholar were employed, as they are widely recognized for bibliographic searches. Scopus and Web of Science are considered reliable data sources due to their extensive coverage of peer-reviewed journals, which include the most recent articles. Google Scholar was used as a complementary source to partially broaden the scope of the previous databases. The search fields were configured as Article Title, Abstract, and Keywords, and several filters were applied during the data collection process. Several combinations of four search strings were developed, while keeping the first one constant (related to glass waste), and using the Boolean operators AND and OR. In this context, the inclusion and exclusion criteria were based on the Preferred Reporting Items for Systematic Reviews and Meta-Analyses (PRISMA) framework.

Regarding the inclusion criteria, the following were considered:Type of study: peer-reviewed journal articles, theses, technical reports, or patents.Time period: publications between 2015 and 2025.Language: studies written in English.Thematic relevance: studies directly related to glass waste, and connection with computer vision.Full-text availability: only articles with complete text available for review.

As for the exclusion criteria, the following aspects were taken into account:Thematic irrelevance: for example, studies dealing with the recycling of materials other than glass, or computer vision applications unrelated to waste management.Duplicates: repeated publications or preliminary versions of the same study.Lack of full-text access: abstracts, posters, or articles not available in full.Language other than English.Low quality or lack of scientific rigor: studies without a clear methodology or with unverifiable results.Publication year outside the defined range.

This approach yielded a total of 287 articles concerning glass. After screening the titles and abstracts, non-relevant works were excluded, resulting in 98 studies directly related to waste management. However, some manuscripts not strictly connected to the topic were also included. Given the narrow scope of this research field, the review ultimately provided a relatively limited number of scientific references.

There are not many scientific works published in high-impact journals or conferences that address computer vision techniques, particularly with respect to sensors and technologies in the glass waste issue. However, solutions, reports, patents, and analyses exist in grey literature. For this reason, we believe that their inclusion significantly enriches the review. All these references have been permanently preserved using the Perma.cc and Internet Archive Wayback Machine platforms. References to companies, patents, and industry documents and reports were selectively included according to the following criteria:The source provides information directly relevant to the topic and presents data not available in peer-reviewed literature.The authoring entity, company, university, or individual is recognized in the field.The information is recent and relevant to the current state of knowledge.The source is openly accessible and cited in full, including the URL of the archived websites or DOI.

To provide clearer information to the reader, in the following sections (Section 4, Section 5, Section 6, Section 7 and Section 8) we have added a set of acronyms as subscripts to each reference to indicate its origin: Peer-reviewed source/Book (A), Vendor-based source (B), Project deliverable (C), and whether the information comes from a scientific journal or conference (1), a patent (2), or a Web page (3). Accordingly, each reference will appear as follows: [XX] (A1).

## 4. Overview of Equipment and Technologies

The increasing diversity of sensing technologies applied to glass waste management requires a structured view to understand their operational principles and application domains. To facilitate this understanding, Figure 3 provides an overview and taxonomy of the main sensing modalities used across the glass recycling chain. This conceptual map links each modality to its underlying physical principle, typical detection target, and location within the plant—serving as a visual guide to Section 5 and Section 6, where these technologies are described in detail. By presenting this taxonomy upfront, readers can better contextualize the subsequent technical discussion and appreciate the complementarity among the different sensing approaches. Additionally, Figure 4 illustrates the main sensors discussed below and their typical placement along the conveyor in a glass sorting line.

## 5. Equipment and Technologies for Glass Waste Management (Excluding Computer Vision)

A typical MSW management cycle usually consists of four main processes: generation, collection, treatment, and disposal [11] (A1). Figure 5 outlines a standard process used for glass recovery and recycling. Aside from beverage glass bottles that are either reused or returned directly to supermarkets, post-consumer glass from households is usually collected through either single-stream systems (in designated containers) or multi-stream systems, where it is combined with other recyclables such as paper, plastics, and metal scrap. Once collected, the glass is commonly sorted using various separation methods at a materials recovery facility and then crushed into cullet fragments measuring approximately 1 to 2 cm. Following the removal of contaminants—such as plastics, metals, ceramics, and off-colour glass—through additional separation steps, the cleaned cullet can be reused as a secondary raw material for manufacturing new glass products or other applications.

The technologies used in glass recycling can be categorized according to the main stages outlined in Figure 5. These categories are summarized as: collection, sorting, and recycling. This section provides a review of techniques excluding computer vision–based methods, which will be addressed separately in the following section.

### 5.1. Technologies for Collecting Glass Waste

The collection of glass waste is a critical stage in the recycling process, as it largely determines the quality of the material that will reach treatment facilities. Collection is still primarily carried out by human operators. In many European countries, the most common method involves the use of dedicated glass containers—typically green—where citizens are instructed to deposit only glass packaging. This selective collection system enables source separation, which helps reduce material contamination and facilitates subsequent recycling.

However, daily collection of solid waste presents certain drawbacks, such as wasted time, fuel, and labour when containers are found to be empty. On the other hand, extending the collection interval increases the risk of containers overflowing, which can result in public health issues and environmental pollution. In addition to urban collection containers, there are designated recycling centres or ecoparks where special or bulky glass items—such as windowpanes, mirrors, or flat glass—are collected separately, as they should not be mixed with packaging glass. In some regions, particularly in densely populated urban areas, door-to-door collection systems are also implemented. However, this approach is less common for glass due to its weight and volume.

Trucks and Robotic Arms

In recent years, the automation of waste collection trucks, including those used for glass, has progressed significantly due to the development of mechanical systems such as side-mounted robotic arms. These devices enable the automatic lifting, emptying, and repositioning of containers, minimizing human intervention while improving safety and operational efficiency. The structural design of these robotic arms takes into account factors such as the weight of the glass, container geometry, and vehicle stability during operation. For instance, Yuan et al. propose an optimized arm structure based on finite element simulation and dynamic analysis, capable of performing precise and repetitive collection tasks without damaging the container or compromising the truck’s balance [38] (A1).

These types of robotic arms are not only deployed in experimental setups but are also actively used in real urban waste collection scenarios, as documented in Japan and Nordic countries. In these contexts, automated glass collection forms part of broader smart city initiatives aimed at reducing operational costs and expanding service coverage. Owaga et al. [39] (A1) present a contactless collection system integrating robotic pickers, IoT sensors, and route optimization using Laser Imaging Detection and Ranging (LIDAR) and Big Data analytics. Similar fully operational systems can be found in Sweden (Volvo-Renova) and China. Other models feature multifunctional arms adaptable to various container types and waste materials, enhancing fleet efficiency [40] (A1).

Weight sensors

Many of the robotic arms mentioned previously are equipped with additional sensors as part of an intelligent control and monitoring system. For instance, weighing sensors (load cells) are installed either on the robotic arm or on the truck platform [20] (A1). These systems enable the measurement of the container’s weight before and after emptying, which is useful for logistical control, volume-based billing, and overload detection. While predominantly used in industrial applications, these sensors are less common than level sensors in logistics contexts [41,42] (A1).

Ultrasonic Sensors

In the glass collection phase, smart containers are equipped with various types of sensors. The most common among these are ultrasonic sensors, typically installed at the top interior of the container to measure fill levels by emitting sound pulses and detecting their echoes, achieving accuracies of up to ±1 cm. The data is transmitted in real-time via IoT networks such as LTE or LoRaWAN [20] (A1). Although this information is limited, it remains useful for operators responsible for emptying the containers.

Temperature Sensors

In some advanced cases, smart containers include temperature and tilt sensors, which provide alerts for unwanted tipping or heat conditions that could compromise the integrity of the contents [20] (A1). These technologies enable better anticipation of collection routes, help avoid unnecessary trips, and optimize the operational efficiency of glass collection fleets.

Sensors and Communication Networks

Technological advances in wireless communications are paving the way for the automation of solid waste management systems [43] (A1). In this field, various approaches are being explored, including RFID-based systems, wireless sensor networks (WSNs), and Internet of Things (IoT)-based technologies.

A typical system architecture for an RFID-based Solid Waste Handling System is composed of communication technologies such as general packet radio service (GPRS), global positioning system (GPS), radio frequency identification (RFID), and geographic information system (GIS) [23] (A1). In this system, an RFID tag is attached to each waste container, and the collection truck is equipped with an RFID reader, a camera, a GPS module, and a GSM module. The RFID reader detects the tag installed on the container, while the GPS module captures real-time location data. All the information collected by the truck is recorded and transmitted to a central monitoring station via a GSM or GPRS network. The monitoring station includes a receiver, a Geographic Information System (GIS), a database, and a user interface terminal. The GIS and database management systems are responsible for mapping the truck’s position and container locations in order to optimize collection routes based on estimated waste volume.

The main limitations of RFID-based systems include the lack of real-time monitoring, restricted communication range, and the requirement for additional infrastructure to effectively track container status.

In approaches based on Wireless Sensor Networks (WSNs) for solid waste management during the collection phase, a Wireless Sensor Network is composed of a large number of wireless sensor nodes deployed ad hoc to monitor physical or environmental parameters—in this case, those related to waste containers. For example, sensor nodes attached to waste bins can measure the fill level and transmit this data to a wireless access point [44] (A1). In a typical WSN, each sensor node is equipped with an embedded CPU to monitor the environment in a specific area. These nodes are connected to a central unit known as the coordinator node, and all coordinator nodes are linked to a base station, which serves as the main processing unit of the WSN.

Compared to RFID-based systems, WSN approaches offer enhanced capabilities; however, they face challenges related to designing networks that are energy-efficient and easy to deploy.

An IoT environment consists of a network of web-connected smart devices that use embedded electronic components—such as communication hardware, sensors, and processing units (CPUs)—to collect, transmit, and process data from their surroundings. By connecting IoT devices to an edge device, such as an IoT gateway, sensor data can be shared and transmitted to the cloud for further analysis. The use of low-power sensors, wireless microcontrollers, and lightweight communication protocols has enhanced IoT-based approaches to solid waste management ([45,46] (A1)). Moreover, these solutions have overcome the limitations of traditional WSN-based systems by offering interoperability and dynamic adaptation of the sensor nodes.

### 5.2. Waste Sorting and Classification Processes

Once collected, glass waste undergoes separation and sorting processes. Waste classification—often used interchangeably with waste separation—can be carried out manually at the source [47] or implemented at a relatively centralized facility [48] (A1).

Magnetic Separators

Magnetic separators are used during the sorting phase of recycled glass (cullet) to remove metallic contaminants that may be mixed with the fragments. Although glass is not a metallic material, this type of separation is essential to ensure its purity before being reprocessed in industrial furnaces. Purity is defined as the proportion (in %) of the recovered output stream that consists exclusively of clean glass, with all non-glass contaminants—such as plastics, metals, ceramics, or paper—being removed [49] (A1). These technologies are mainly applied in treatment plants for glass packaging waste, rather than at the household collection stage or in urban container systems. Several technologies are specifically discussed in the following paragraphs

Overband magnetic separators are installed above conveyor belts to remove large ferrous objects such as nails, caps, or metal sheets that may be mixed with the glass. Magnetic separation systems play a crucial role in processing crushed glass (cullet), as they eliminate metallic contaminants and ensure a higher-quality recycled glass product. These systems use magnetic fields to attract and separate ferrous (magnetic) metals and, in some configurations, also non-ferrous (non-magnetic) metals from the cullet stream.

In the 1960s, Arutsev [50] (A1) investigated the use of magnetic separators to remove the “crown skin”—a contaminated layer of oxides—from cullet in the glass industry. This pioneering study demonstrated that magnets positioned over conveyor belts are effective at removing ferrous metals adhered to crushed glass, thereby improving the quality of the processed material.

More recently, Bellopede et al. [4] (A1) presented an analysis conducted at a plant in northern Italy that combines screening, density separation, and manual magnetic separation using 1 Tesla magnets on cullet. This process allowed for the removal of significant amounts of ferrous and non-ferrous metals before optical systems were applied, reducing impurities and notably improving the quality of the recycled glass.

Magnetic pulleys and magnetic drums enable a finer separation of metal particles, thanks to high-intensity magnetic fields generated by permanent magnets [51] (B3). These systems are usually positioned at the beginning of the sorting process.

Finally, eddy current separators remove non-ferrous metals such as aluminium, brass, or copper. These contaminants often originate from fragments of bottle caps or closures accidentally mixed with the glass [24] (B3).

Air suction systems

Air suction systems also play a crucial role in cleaning glass cullet in recycling plants by removing lighter materials such as paper, plastics, labels, and organic residues. In [25] (B3), a zigzag airflow configuration, is used to effectively separate these contaminants from the glass, a method widely adopted in the glass container recycling industry to improve material purity. Additionally, companies like Binder + Co [52] (B3) incorporate systems known as “Breezer,” which use blown air to remove fibres and lightweight elements immediately after glass screening and grinding processes. Sometimes, these systems are part of combined sorting units, such as air classifiers or air knives, which use controlled air currents to lift and separate light materials, allowing clean glass to fall, thereby enhancing operational efficiency and reducing the workload of subsequent optical sorters [53] (B3).

In a study conducted by Bellopede et al. [4] (A1) at an Italian glass recycling plant, the use of air classifiers to separate fine fractions from the main stream was documented, achieving a reduction in cullet impurity from less than 25% to approximately 12.7% in the <1 mm fraction. These technologies are typically integrated after crushing and screening, and before more advanced optical separation stages, significantly contributing to the improvement of cullet purity and the energy efficiency of the melting processes.

Screens and Crushers

In the classification phase of recycled glass, vibrating screens separate cullet into specific particle size fractions, which facilitates subsequent treatment through optical or magnetic separation. For example, a recent study analysed the use of vibrating feeders designed to remove light impurities (such as labels or plastic pieces) from crushed glass containers, detailing the design of these systems to maximize efficiency in extracting unwanted fines [26] (A1). These systems are usually positioned after crushers such as roll crushers or VSI impact crushers, which break the glass down to sizes smaller than 10 mm, allowing the screens to evenly distribute the material and optimize the performance of subsequent cleaning stages [54] (A1).

In conclusion, it can be stated that effective screening not only enhances the efficiency of subsequent technologies, such as colour or density separation, but also reduces equipment wear and optimizes process traceability in automated lines. Although mechanical in nature, these solutions form an essential foundation for the integration of more advanced technologies like computer vision or spectral sensors.

### 5.3. Representative Studies

Table 2 provides a summary of representative references for each of the technologies covered in this section. In addition to details about authorship and publication type, it shows the context in which the glass waste was generated, which other materials alongside glass were addressed in the study, the stage of the glass waste treatment process in which the study is situated, and the ultimate objective of the work.

## 6. Computer Vision-Based Technologies for Glass Waste Management

In this section, which forms the core of the present article, the main computer vision technologies applied in some way to glass waste are addressed.

Glass waste collection at the source and the classification of waste at sorting and recycling facilities are the two primary scenarios where computer vision systems are implemented. Approximately one in four publications focus on the first scenario [19] (A1). In that case, the systems are typically limited to CCD cameras or stereo vision setups. In contrast, the second scenario features a much broader range of image-based technologies, used primarily for sorting and separating glass materials. The performance of glass identification and classification systems can be evaluated using standard classification metrics such as accuracy (overall correctness), precision (the proportion of correctly detected glass instances among all predicted as glass), and recall (the proportion of actual glass instances that were successfully detected) [55] (A1).

The following sections provide a detailed overview of how current technologies are used in both environments.

### 6.1. Computer Vision in Glass Waste Collection Tasks

RGB Cameras

The use of simple 2D cameras (CCD or CMOS) or stereoscopic cameras is common in various tasks during the glass collection phase. For instance, RGB cameras can be installed to provide an environmental view and monitor any circumstances that might hinder the normal operation of containers or articulated arms in unstructured environments [56]. The application of these sensors for monitoring purposes is currently expanding, particularly within pilot projects related to smart cities [40] (A1).

Although vision technologies have been applied on the exterior of glass containers, there is currently limited scientific literature specifically using 2D cameras inside them. However, there are “smart bin” projects where 2D cameras (standard RGB) are used to detect, classify, and visually monitor waste before it is deposited. For example, Wahyutama and Hwang [57] (A1) developed a smart bin that uses a webcam (2D) to classify packaging waste. Another automatic detection approach is that of Bansal et al. [58] (B3), where 2D cameras mounted on a robotic arm are used. Using AI techniques, the system identifies waste and calculates its position inside the container, achieving over 95% real-time recognition. In both cases, 2D computer vision is shown to be effective in identifying, guiding, and managing waste in automated containers handling glass.

Nowakowski y Pamuła [59] developed a system based on convolutional neural networks (CNN) and R-CNN to classify e-waste using images captured with mobile phones. Among the studied categories, glass is included as a component in electronic devices such as screens, being one of six materials classified along with copper, PCBs, aluminium, steel, and plastic. The accuracy rates obtained ranged between 90% and 97%, depending on the waste category, demonstrating robust performance even for challenging materials like glass. The main goal of this system is to facilitate the exchange of information between citizens and waste collection companies.

There are more examples of smart bins where imaging resources have been widely adopted in smart cities and campuses worldwide [60] (A1). These bins feature a compaction mechanism that increases their capacity, along with automated real-time collection notifications. Although several authors have conducted research on so-called smart bins—intelligent containers integrating computer vision and sensors to detect fill levels or even perform preliminary automatic waste sorting ([61,62,63,64] (A1)), none of the analysed works specifically focus on glass waste.

Research by Hannan et al. [23] (A1) and Aziz et al. [62] (A1), which involves the use of machine learning techniques or Hough line detection to estimate bin fill levels, and the study by Jacobsen et al. [65] (A1), which explores the integration of computer vision with robotics to sort waste inside the container itself, mainly concentrate on general waste or preliminary sorting without explicitly addressing glass separation as an individual material. This highlights a significant gap in current literature since glass, despite being a common and recyclable waste, has yet to be thoroughly explored in the context of smart bins.

One of the few systems that includes glass as an essential component is the work by White et al. [66] (A1). The authors propose WasteNet, a waste classification model based on convolutional neural networks that can be implemented on a low-power edge device (i.e., data processing performed directly on the local device where the data is generated), such as a Jetson Nano. Automated waste classification at the edge enables smart and fast decision-making in smart bins without the need for cloud access. In this study, the waste is classified into six categories, including glass alongside paper, cardboard, metal, plastic, and others. The model achieves a prediction accuracy of 97% on the test dataset.

LIDAR and Stereo cameras

LiDAR is widely used for mapping, navigation, and object detection, but glass presents unique challenges due to its transparency and reflective properties. Recent research has focused on improving LiDAR’s ability to detect, map, and reconstruct glass structures, which is crucial for robotics, autonomous vehicles for glass collection, and waste management in environments with significant glass content ([67,68] (A1)).

Using stereoscopic cameras, it is possible to verify the correct positioning of the container on the ground, detect obstacles, and document any incidents that may occur (e.g., broken container, blocked access, etc.) [30] (A1). Similarly, in smart waste collection, including glass waste, 3D vision systems based on RGB-D cameras or LiDAR technologies are increasingly used to improve the detection and precise localization of objects inside containers. For example, Páez Ubieta et al. [56] (A1) developed a mobile system combining an RGB-D camera with LiDAR to estimate waste depth and calculate its position in outdoor environments, achieving localization accuracy on the order of centimetres. Likewise, Zhang et al. [69] (A1) present platforms equipped with 2D LiDAR and RGB cameras mounted on waste collection vehicles (WCV), which enable obstacle detection, container identification for emptying, and real-time environment mapping. Together, these technologies demonstrate that 3D vision systems provide significant advantages in accuracy, safety, and efficiency for automated waste collection.

### 6.2. Computer Vision in Glass Waste Sorting and Classification

RGB cameras

In modern glass recycling plants, RGB (2D) cameras play a key role in the optical sorting of cullet by enabling the identification of different glass colours (clear, amber, green) and the detection of contaminants such as ceramics, labels, or light metals. These cameras are integrated into machine vision systems that operate in real time and are connected to compressed air nozzles or automated arms for ejection.

A prominent example in this field is the SEEGLASS system, developed under the European Horizon 2020 program [27] (C1), which uses RGB cameras combined with machine vision algorithms to distinguish glass by colour and detect impurities. This system achieved a purity of 99% and a recovery rate of 85% of the glass contained in municipal waste. Recovery rate is defined as the percentage of the total glass input that is successfully recovered as usable cullet after the sorting and cleaning processes, excluding the fraction lost due to contamination or separation inefficiencies. Similar technologies have been described in patents such as EP1752228B1, where RGB and NIR cameras allow classification by shade and material type, automatically triggering rejection or selection mechanisms [70] (A2). At the industrial level, solutions like the company PICVISA [28] (B3) use visible spectrum (VIS) cameras along with neural networks to differentiate between glass and other materials, adjusting classification by colour and shape.

From the scientific field, Krcmarik et al. [71] (A1) developed an industrial vision system for glass classification based on an RGB line-scan camera and FPGA processing, achieving accurate detection of impurities adhered to glass under harsh industrial conditions. On the other hand, Cheng et al. [72] (A1) designed an automated system using RGB cameras and deep learning algorithms that includes the classification of glass bottles within mixed waste. They employ neural networks for their identification and separation using a robotic manipulator. Echeverry et al. [73] (A1) proposed an automated sorting system based on computer vision, which enables the recognition and separation of recyclable materials (plastic, glass, cardboard, and metal) through a webcam connected in real time to the Nvidia^®^ Jetson Nano™.

The aforementioned studies and industrial developments demonstrate how RGB cameras have become a key tool for improving the efficiency and purity of recycled glass, reducing manual intervention and optimizing automation in waste treatment plants. However, they face a significant limitation: the visual similarity between glass and other materials, such as transparent plastics, makes accurate identification using only the visible spectrum (i.e., RGB) difficult. Therefore, it is necessary to complement these cameras with other types of sensors to achieve more precise classification.

Hyperspectral or Spectral Cameras (HSI)

Whereas RGB cameras are based on the visible spectrum and capture light in three broad bands—B (blue): ~400–500 nm, G (green): ~500–600 nm, and R (red): ~600–700 nm—hyperspectral or spectral cameras capture information across dozens or even hundreds of bands in the electromagnetic spectrum. These include the visible, near-infrared (NIR, 700 to 1000 nm), and sometimes short-wave infrared (SWIR, 1000 to 2500 nm), which are parts of the non-thermal infrared spectrum.

Hyperspectral imaging (HSI) is an emerging technology with promising applications in waste identification. HSI offers a potential advantage over NIR systems by integrating both spectroscopic and visible imaging capabilities into a single system [74] (A1). This enables the simultaneous acquisition of both spectral and spatial information. However, HSI images are expensive due to the complexity of the equipment required to capture a broad continuous spectrum; at the same time, they are more robust to sunlight reflections, which represent a major limitation for RGB cameras.

Glass is transparent in the NIR range, and therefore, does not reflect or absorb much in those wavelengths. However, certain bands in the SWIR range can show characteristic absorptions depending on the type of glass (soda-lime, borosilicate, quartz), impurities (metal oxides), presence of water (surface hydration), and thermal or chemical treatments. Bonifazi et al. [31] (A1) introduced a novel method for distinguishing ceramic glass using imaging spectroscopy. Reflectance spectra were acquired with two different instruments covering the visible and near-infrared ranges (400–1000 nm and 1000–1700 nm). The findings demonstrated that both standard glass and ceramic glass can be effectively identified by analysing specific wavelength ratios within these spectral regions. Similarly, the industrial project WiserSort, developed by Lenz Instruments [75] (B3), integrates hyperspectral cameras with deep neural networks to classify materials based on their spectral signature, including glass fragments within packaging waste. Additionally, a recent review by Menezes et al. [76] (A1) highlights that HSI systems installed on conveyor belts, combined with artificial intelligence algorithms, have achieved identification of glass, plastics, and metals with over 95% accuracy in industrial settings.

Infrared Cameras (IR)

Following Planck’s law, infrared cameras detect infrared radiation emitted by hot objects. The typical range for these cameras is the mid-infrared (MIR) or far-infrared (FIR) spectrum, from approximately 3000 to 14,000 nm, and they are commonly used to measure temperature or detect heat.

Thermal or far-infrared (FIR) cameras are being explored for glass segmentation and detection in waste streams, taking advantage of glass’s distinct thermal transmission compared to other materials. Huo et al. [32] (A1) developed a method for glass segmentation using paired RGB and thermal images, leveraging the fact that glass is transparent in the visible spectrum but opaque in the thermal spectrum. This allows reliable segmentation through multimodal neural networks. Their approach, which uses a set of RGB and thermal images combined with a CNN-transformer architecture, demonstrated high effectiveness in distinguishing glass regions in varied scenes. Although this model has not yet been directly applied in recycling plants, it represents a significant step forward toward automatic glass detection systems in mixed environments. Additionally, previous research on thermal vision for material classification, such as the system developed by Cho et al. [77] (A1), shows that various materials (including glass) can be recognized with high accuracy (>98%) using mobile thermal cameras and deep learning, paving the way for promising applications in smart bins or automated sorting lines.

Stereo RGB Cameras

Stereo or multi-view (3D) vision systems [78] (A1) are primarily used in waste sorting to improve the accuracy of object detection and localization, and they have recently begun to be applied to glass as well. An illustrative example is the work by Wu et al. [29] (A1), which employs a binocular stereo vision system to characterize multiple recyclable objects in household environments. Although it is not exclusively focused on glass, the same principle can be applied to discriminate glass fragments in mixed waste streams, achieving distance accuracy better than 5% within ranges of 0.5 to 1.2 m. Similarly, Oh et al. [30] (A1) combined stereo cameras to automate the selection of objects inside containers by precisely estimating their three-dimensional position in sorting environments. These technologies can be integrated into cullet sorting lines, enhancing glass purity by identifying and removing solid contaminants or unwanted fragments.

Active 3D Vision Systems

Active 3D vision systems, such as structured-light cameras [79] (A1), LiDAR, or time-of-flight (ToF) sensors, provide superior spatial detection capabilities that can enhance the sorting of glass cullet, especially when dealing with irregular shapes or stacked fragments. In industrial settings, 3D structured-light cameras mounted on robotic arms can generate precise point clouds of transparent or shiny objects like glass fragments, with capture speeds under 150 ms per frame—facilitating their integration into automated collection lines [80] (B3). Additionally, recycling research has examined the use of structured light combined with HSI sensors to distinguish dark or contaminated glass, as seen in systems like Sesotec SPEKTRUM, which enhance the identification of black glass through active light patterns paired with spectral detection [33] (B3). These technologies demonstrate the potential of active 3D vision to improve the accuracy and consistency of glass sorting, reducing false negatives in complex mixed waste environments and boosting separation efficiency.

Polarimetry in Computer Vision

Polarimetry in computer vision [81] (A1) is an emerging technology that leverages the polarization properties of light reflected and transmitted by glass to distinguish it from other materials. It takes advantage of how glass, due to its dielectric nature, alters the polarization state of reflected light. Although this technology is still not widely covered in the literature, it has proven useful, for example, in differentiating glass from transparent plastics.

The precise identification of transparent objects, such as those made of glass, is essential for multiple applications, including the safe operation of autonomous vehicles, assisted vision, and automated navigation systems. In the recycling field, recognizing transparent objects is crucial for sorting waste and items, accurately reconstructing glass materials, and, in industrial contexts, detecting manufacturing defects in glass substrates used for displays. Quan et al. [82] (A1) provide a review that highlights the importance of recognizing and measuring transparent objects. Meanwhile, Taglione et al. [34] (A1) examine current applications of polarimetry for detecting transparent objects like glass in real-world environments.

Additionally, passive polarimetry techniques have been used to segment transparent objects in urban or robotic environments, where glass can be detected more reliably due to its distinct polarimetric behaviour compared to opaque solids [83] (A1). Other broader studies on material classification using polarimetry confirm that glass, as a dielectric, can be clearly distinguished from metals and transparent plastics by its unique polarization signature [84] (A1). These findings provide a strong foundation for the future use of polarimetry in cullet sorting lines, especially to improve the detection of transparent glass, whether stacked or mixed with other waste.

Fluorescence Vision

Although still an emerging technology in glass recycling, fluorescence or UV vision can exploit the fluorescence properties exhibited by certain types of glass when excited with ultraviolet light. The patent by Reinhold et al. [35] (A2) describes an optical system for sorting recycled glass using UV light excitation and analysis of the emitted fluorescence (in the UV–VIS range), which allows the identification of contaminants such as lead found in some glass containers. This method proves to be cost-effective and reliable for online sorting of special glass, regardless of the material’s colour or shape. This type of technology, based on spectral sensors combined with UV illumination, offers a potential approach for classifying glass types according to their composition or detecting contaminated glass in mixed waste streams, although its implementation in commercial industrial plants has not yet been documented.

Active Laser Spectroscopy and Optical Computed Tomography

These techniques are employed in laboratory settings to characterize micro-defects or the thickness of glass, with potential applications in technical glass recycling, such as for screens and flat glass.

Active laser spectroscopy techniques, particularly Laser-Induced Breakdown Spectroscopy (LIBS) [85] (A1), have recently been explored for characterizing glass cullet prior to remelting. A study published in the *Journal of Non-Crystalline Solids* (2024) demonstrated that using LIBS combined with machine learning—specifically support vector machines (SVM)—enables distinguishing glass types by colour (e.g., flint and green) as well as differences in particle size, achieving over 99% accuracy in colour identification and 97% in cross-identification of type and granulometry [36] (A1). Although LIBS does not reconstruct 3D shapes, laser tomography via LIBS provides spectral and chemical information, making it an advanced technique for pre-classifying and ensuring glass quality before subsequent optical processing stages.

Regarding Optical Coherence Tomography (OCT), it is an emerging technique that enables volumetric imaging of transparent or translucent objects with micrometre resolution [86] (A1). While its current applications focus mainly on medical and industrial fields—such as thickness measurement and defect detection in semiconductors or technical glass—its direct use in recycling lines for containers has not yet been reported. However, OCT holds future potential for detecting microcracks or discontinuities in glass fragments intended for technical recycling.

Hybrid Imaging Technologies

Hybrid computer vision technologies for glass classification combine data from multiple sensors—such as RGB cameras, 3D systems (LiDAR, structured-light), and spectral sensors (hyperspectral or multispectral)—to enhance accuracy and robustness in complex environments. By integrating different sensing modalities, these systems achieve a more comprehensive understanding of glass waste streams. However, incorporating such multi-sensor setups into existing infrastructure increases costs and poses challenges related to calibration, data synchronization, and processing complexity.

In industrial waste-sorting environments, multimodal sensor fusion—combining RGB data, spectral information (HSI), and in some cases, depth or induction data—has been shown to significantly improve the detection of transparent materials, including glass. A representative example is the SpectralWaste dataset, studied by Casao et al. [37] (A1), which was acquired in a real recycling facility using synchronized line-scan RGB cameras and hyperspectral sensors to segment and classify objects in real time. Results indicate that spectral + RGB integration outperforms systems relying solely on conventional vision, especially when dealing with translucent materials.

Along similar lines, Arbash et al. [87] (A1) introduced the Electrolyzers-HSI dataset, which includes both hyperspectral and RGB images for material classification in battery recycling. Their approach leverages deep learning architectures such as Vision Transformer and Multimodal Fusion Transformer to improve material identification accuracy. In Tao et al. [88] (A1) hyperspectral imaging (VIS–NIR) is combined with machine learning algorithms and further enriched by integrating depth or laser texture parameters from real samples of urban waste. The system achieved classification accuracy above 97% for glass (including clear glass) versus plastics and paper. There are other strategies based on the combination of imaging sensors that could provide significant results in the future. For example, the fusion of depth-sensing technologies with colour information would make it possible to estimate the volume of glass waste based on previously developed silhouette estimation algorithms [89] (A1).

In conclusion, it can be asserted that data fusion enables the overcoming of individual limitations inherent to each imaging technology—for instance, the difficulty of distinguishing dark glass using RGB alone, or the inability to detect contaminants using only 3D sensors. These hybrid solutions are becoming increasingly common in pilot projects for smart cities and advanced recycling facilities.

### 6.3. Representative Studies

As in Section 5, this section provides a summarized overview, in Table 3, of the most representative publications for each computer vision-based technology discussed. In addition to details about authorship and type of publication, the summary includes information on the environment where the glass waste was generated, whether other materials were present alongside the glass in the study, the specific stage of the glass waste treatment process in which the study is situated, and the ultimate objective of the work.

## 7. Brief Remarks on Algorithms and Public Image Datasets for Glass Waste

Although the focus of this review is exclusively on imaging sensor technologies for glass waste treatment, it is worth briefly mentioning current methods and algorithms applied to glass images, as well as some of the datasets available in this field.

### 7.1. Computer Vision Algorithms

The predominant computer vision (CV) algorithms for glass waste classification can be grouped into three main categories: classical algorithms, traditional machine learning (ML) approaches, and advanced deep learning (DL) techniques [90] (A1).

When it comes to classic methods (excluding machine learning and deep learning), glass classification in images typically relies on conventional image processing and computer vision techniques. Common approaches involve edge detection algorithms—such as Canny or Sobel—to identify object contours, as well as texture analysis to differentiate glass from other materials based on surface characteristics. Colour and intensity thresholding are also employed to leverage the transparency and reflective properties of glass. Additionally, features related to reflection and refraction are used, as glass often produces distinctive highlights and distortions in images. Some methods incorporate multispectral or multimodal data, combining, for example, visible and thermal imagery. This is particularly effective because glass is transparent in the visible spectrum but opaque to thermal radiation, allowing for straightforward segmentation using thresholding or subtraction techniques in thermal images [32] (A1). In addition, techniques based on object shape and geometric feature extraction are used to classify glass by analysing contours, symmetry, and structural patterns. Although these traditional approaches offer advantages in terms of speed and interpretability, they often face challenges in real-world conditions—such as cluttered backgrounds, inconsistent lighting, and the intrinsic transparency of glass. These limitations have led to a growing interest in machine learning and deep learning models, which offer greater accuracy and resilience in complex scenarios. Other studies have proposed classification algorithms for glass bottles based on well-known Scale-Invariant Feature Transform (SIFT) descriptors and Haar-Like Shape Network (HLSN) functions, which are similar to Haar-like features implemented through Spatial Pyramid Pooling Network (SPP-Net) architectures [91] (A1). Others propose a pattern recognition method designed to discriminate between objects made of plastic, metal, or glass, based on orthogonal wavelet decomposition techniques [92] (A1).

Traditional machine learning algorithms—including Random Forest, Support Vector Machines (SVM), K-Nearest Neighbours (KNN), Logistic Regression, and Gaussian Mixture Models—have been extensively applied in glass classification tasks. These approaches often utilize features extracted from either the chemical and physical characteristics of glass or directly from image data. Random Forest and SVM, in particular, tend to deliver strong performance, especially when fine-tuned using optimization strategies such as grid search or Bayesian optimization [93] (A1). Such methods have proven effective in accurately distinguishing transparent materials.

Machine learning algorithms require manually labelled visual features as input, whereas deep learning algorithms can automatically extract hidden features directly from raw images. Due to their advantages in robustness and end-to-end automated training, deep learning has become the predominant computer vision approach for facilitating the classification of solid glass waste ([94,95,96] (A1)). Deep learning techniques, such as Artificial Neural Networks (ANN), Convolutional Neural Networks (CNN), Recurrent Neural Networks (RNN), Bidirectional Long Short-Term Memory (BiLSTM), and deep neuro-fuzzy networks, have demonstrated excellent results in both identifying different types of glass and detecting glass areas or defects within images. Some of these models have reached accuracy rates exceeding 90%. For tasks involving image-based glass detection and segmentation, cutting-edge methods utilize advanced architectures including transformers, attention mechanisms, and multimodal fusion combining RGB, depth, and thermal data to overcome the challenges related to glass’s transparency and reflective properties.

With the widespread adoption of ML and DL, there has been a tendency to simplify the problem of waste recognition within certain system constraints; for example, some authors assume that waste items are placed on the conveyor belt one by one [97] (A1) or that they do not overlap. Recent DL studies typically focus on recognizing individual objects from images [98] (A1). Such simplifications, involving strict limitations, may render the developed approaches incompatible with real-world, unstructured sorting environments where waste is usually scattered randomly over cluttered backgrounds ([99,100] (A1)). New insights derived from earlier computer vision methods could be applied to the sorting line stage. For instance, as an initial step towards enabling interaction between industrial robots and dynamic environments in glass waste facilities, a colour structured-light technique inspired by the disordered codeword pattern proposed in [101] (A1) could provide real-time 3D coordinates of glass bottles, thereby facilitating the tracking process.

### 7.2. Image Datasets

There are very few publicly available datasets specifically focused on industrial practices for glass waste sorting. This scarcity has hindered rigorous comparisons between different approaches. Most datasets that include glass are privately owned by their respective research teams. Although some open-access datasets exist, many tend to oversimplify the industrial needs by framing the problem as waste recognition on flat, well-controlled backgrounds. Some studies have started collecting images of waste in real-world contexts, but these remain limited in number. Notable examples include datasets like TrashNet [102] (A1), TACO [103] (A1), AquaTrash [104] (A1), and DataCluster Labs [105] (B3), which contain images of various types of waste such as paper, plastic, metal, and trash and include glass as well, and on which deep learning and machine learning techniques have been applied.

One of the leading image datasets tailored specifically for glass classification in computer vision is the Glass Detection Dataset (GDD) [106] (A1). It was created to tackle the challenge of identifying glass surfaces in images—such as windows or glass doors—that are frequently missed by conventional vision systems. The GDD is a large-scale collection featuring a wide variety of real-world images with annotated glass regions, facilitating the training and evaluation of deep learning models focused on glass detection and segmentation. While general-purpose image datasets like ImageNet [107] (A1), Caltech-101 [108] (A1), and CIFAR are popular in image classification research, they lack dedicated glass categories and thus are not ideal for specialized glass classification tasks. In scenarios where glass classification forms part of broader industrial inspection or surface defect detection, researchers often rely on custom datasets or enhance existing collections with synthetic images generated through methods like Generative Adversarial Networks (GANs) to overcome limited data availability and class imbalance issues ([109,110] (A1)). Nevertheless, for applications explicitly focused on glass identification or segmentation, the GDD remains the most pertinent and commonly used dataset in recent computer vision studies.

In order to complement this section, Table 4 presents publicly available datasets that include glass in realistic settings—beyond clean-background conditions—and that could be applied in industrial environments. These datasets are organized into six categories: Packaging and Containers (T1), Tableware and Kitchenware (T2), Household and Decorative Items (T3), Construction and Architecture (T4), Technical and Industrial Glass (T5), and Automotive and Transport (T6). For each dataset, we report the approximate dataset size, the estimated percentage range of images corresponding to each category, the institution or authors responsible for the dataset, and a brief description of the most common objects included in the image collection.

## 8. Statistical Analysis of Technology Use

Based on the literature reviewed in this paper, Figure 6—consistent with Figure 3—presents the relative frequency of use of each technology, together with the percentage distribution of plant location types for both non-vision and vision-based technologies. A discussion of the results obtained is presented below.

The analysis of non-vision sensing technologies reveals a clear predominance of Magnetic and Eddy-Current systems (27.3%) and RFID/IoT/WSN solutions (22.7%), indicating the strong importance of material identification and automated monitoring in current glass waste treatment facilities. These are followed by Air Suction Systems (18.2%), commonly used for the removal of lightweight contaminants prior to optical sorting stages. Weight sensors still play a relevant role (13.6%), particularly in early bulk flow control or material dosing operations. In contrast, technologies such as Screens & Crushers (9.1%), and especially Ultrasonic and Temperature/Tilt Sensors (both 4.5%), appear less frequently, suggesting that their use is more task-specific and limited to niche process conditions. Overall, the results highlight a strong focus on high-reliability material discrimination and real-time monitoring, rather than general physical measurement or contact-based control.

The distribution of vision-based technologies shows a strong predominance of RGB cameras (36.2%), confirming their role as the most accessible and industry-adopted solution for glass waste inspection, particularly in tasks involving colour-based sorting and shape recognition. Stereo/RGB-D systems follow with 17.0%, reflecting the growing interest in depth-enhanced perception for 3D object discrimination and volume estimation. Emerging polarimetric imaging (10.6%) and hyperspectral/multispectral sensing (8.5%) are gaining relevance, especially due to their ability to distinguish materials based on refractive or spectral signatures rather than purely visual appearance. Hybrid/sensor fusion approaches (8.5%) also indicate a clear trend toward multimodal robustness. In contrast, infrared/thermal sensing (4.3%), active 3D vision and laser spectroscopy/OCT (both 6.4%), and particularly fluorescence/UV imaging (2.1%), appear still limited to more specialized or experimental setups. Overall, these results reveal a transition from conventional 2D imaging toward richer spectral, polarization, and depth-aware modalities, albeit at varying levels of industrial maturity.

The results of this section reveal a clear predominance of traditional RGB-based and electromagnetic sensing modalities, while more advanced spectral, polarimetric, and hybrid configurations are gaining traction but remain less frequently implemented in current industrial environments.

## 9. Discussion

### 9.1. Conclusion and Limitations of Computer Vision Technologies

Although glass is a widely used material in society, industry, and construction, there are no comprehensive reviews on acquisition and classification technologies for glass using computer vision systems. Studying glass acquisition and recycling is complex because most of the existing literature deals with waste where glass is only one component among many. Thus, dedicated reviews focused specifically on glass waste as the central component of the residue are currently non-existent. Therefore, this article poses a challenge within the general literature on waste and recycling from a technological perspective. Algorithms and image processing methods for detecting or classifying glass will be addressed in a second review that will complement the present article.

This article has demonstrated how current computer vision-based technologies can automate complex sorting tasks, reduce operational costs, and improve the quality of recovered glass materials, thereby contributing to the circular economy. The effective use of these CV technologies can also lead to a better understanding of the composition of glass waste, guiding future regulations and aligning industry practices with sustainability standards.

Furthermore, studies have shown that automated computer vision techniques significantly reduce processing times and enhance efficiency compared to manual methods, particularly in the acquisition and classification of glass waste. These solutions enable substantial improvements in processing speed, accuracy in visual identification, and a reduction in dependence on human labour, resulting in significant savings in personnel and operational costs. In real-world environments, accuracy rates above 95% have been reported ([95,119] (A1)), demonstrating their practical applicability and reliability under uncontrolled conditions. Certain experiments have shown that some edge computing platforms, operating without cloud server support, have achieved processing speeds of 23.3 frames per second (fps) with a recognition accuracy of 90.81% [120] (A1). Despite these advancements, there is still room to optimize response times, particularly for real-time applications.

However, the implementation of computer vision-based systems is not without limitations. Among the main challenges are the high installation and maintenance costs of certain imaging systems, as well as the energy consumption associated with continuous operation in real-world environments. Additionally, there is sometimes resistance to adopting these technologies from both users and operators, who may be reluctant to embrace technological change and the integration of new automated methods. Furthermore, aspects such as system robustness remain areas requiring future improvement.

In terms of effectiveness and robustness, many existing classification frameworks show significant shortcomings in accuracy, mainly due to limitations in the feature extraction processes for glass components. Often, these methods fail to produce sufficiently detailed representations of images containing glass or regions of interest, which negatively impacts the results [121] (A1). Note that, while computer vision can be effective in visually identifying a wide range of materials, its capabilities become limited when distinguishing between physicochemical properties is required. This is because materials with very different compositions can share similar visual features. For example, a sheet of transparent plastic may appear almost indistinguishable from glass, despite having entirely different physical and chemical characteristics. In such cases, relying solely on the visual information provided by CV compromises classification accuracy, making it necessary to incorporate complementary or hybrid technologies that provide additional data about the material’s nature.

### 9.2. Challenges and Future Directions

At a conceptual level, the most significant challenges in the automatic identification of waste stem from several factors. First and foremost, there is a notable lack of labelled datasets specifically focused on glass, as well as a shortage of precise and standardized categorizations for this material [122] (A1), which hampers research and development efforts. Therefore, the creation of such labelled image datasets is urgently needed to advance the field.

Another important aspect relates to the complexity of real-world environments and the necessity of improving the current techniques and algorithms. Note that glass is often mixed with various other components in images, making it visually difficult to recognize. Even in scenarios where only glass is present, the wide variation in the shape, size, colour, and transparency of glass materials poses a significant challenge—even for advanced deep learning algorithms such as CNNs. As a result, these AI models may struggle to distinguish between glass bottles and plastic ones due to their visual similarity [123] (A1).

Although there are proposals of smart bins for general waste ([124,125] (A1)), there is a clear gap when it comes to vision-based automatic control solutions during the glass waste disposal phase by users at collection points. While a few semi-automatic systems exist, they offer minimal functionality. For instance, there is a lack of solutions capable of automatically detecting and rejecting incorrectly deposited materials in glass containers or systems that actively encourage user collaboration when disposing of glass packaging. The literature also lacks references to systems that classify the type and properties of deposited glass containers or that measure weight and volume during disposal using computer vision techniques. Only a few low-cost, locally developed prototypes have been reported, typically in conference papers of limited scientific impact.

On the other hand, there is a notable lack of studies addressing human acceptance of computer vision technologies when it comes to managing their own glass waste. In [47] (A1), the motivations behind why people choose to separate their waste and what drives them to follow through in practice are analysed. The study examines how factors such as personal attitude, social pressure, perceived control, moral obligation, and knowledge about recycling influence the intention to sort waste. Based on a survey conducted among citizens in China, the findings show that although many people express a willingness to recycle, this intention does not always lead to action. Moreover, the study highlights that providing information and offering incentives can help individuals move from intention to actual behaviour.

Looking ahead, the development of computer vision-based systems for glass acquisition, identification, and classification is expected to continue expanding—particularly in regions such as the European Union, the United States, and China. Future research should focus on validating these systems in both controlled and uncontrolled environments and integrating multiple sensors by leveraging the benefits of sensor fusion. These efforts will help pave the way toward more robust, adaptive, and efficient systems for automated glass recycling, contributing to broader goals in environmental sustainability.

## Figures and Tables

**Figure 1 sensors-25-06634-f001:**
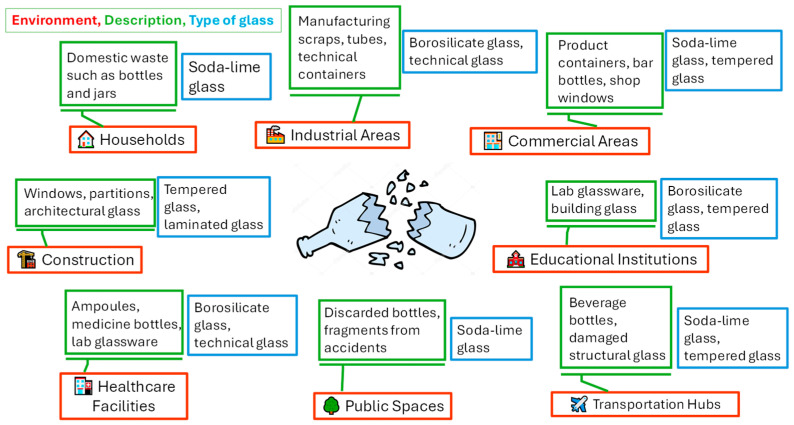
Main sources or environments of glass waste.

**Figure 2 sensors-25-06634-f002:**
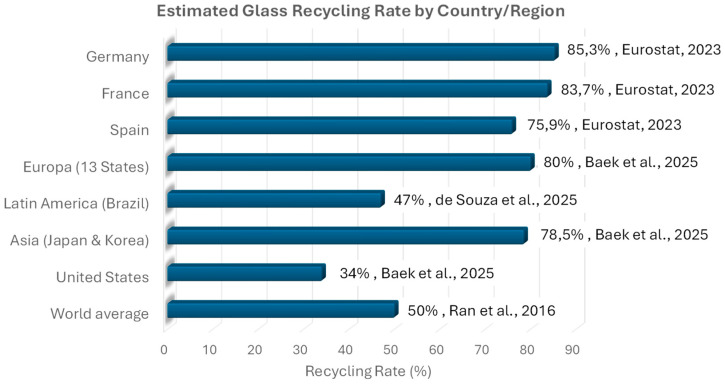
Estimated Glass Recycling Rates in Representative Countries and Continents. Next to each bar are the recycling percentages, along with their corresponding sources and years (Ran et al. [10], Baek et al. [11], Eurostat [12], de Souza et al. [14]).

**Figure 3 sensors-25-06634-f003:**
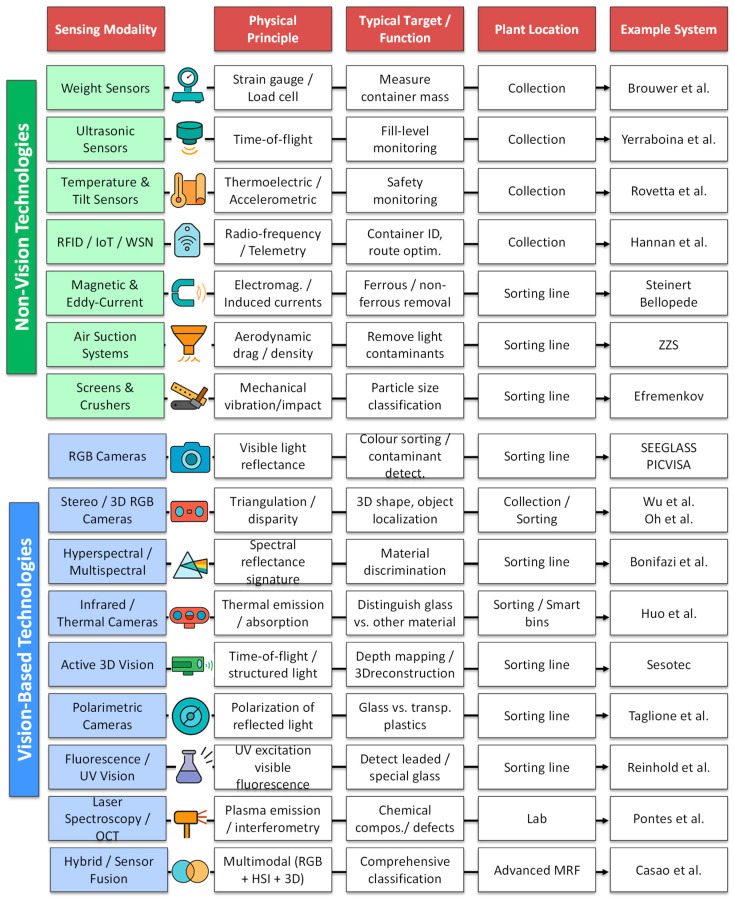
Taxonomy of sensing modalities for glass waste management (Section 5 and Section 6). References are: Brouwer et al. [20], Yerrabiona et al. [21], Rovetta et al. [22], Hannan et al. [23], Steiner et al. [24], Bellopede et al. [4], ZZS [25], Efremenkov et al. [26], SEEGLASS [27], PICVISA [28], Wu et al. [29], Oh et al. [30], Bonifaci et al. [31], Huo et al. [32], Sesotec [33], Taglione et al. [34], Reinhold et al. [35], Pontes et al. [36], Casao et al. [37].

**Figure 4 sensors-25-06634-f004:**
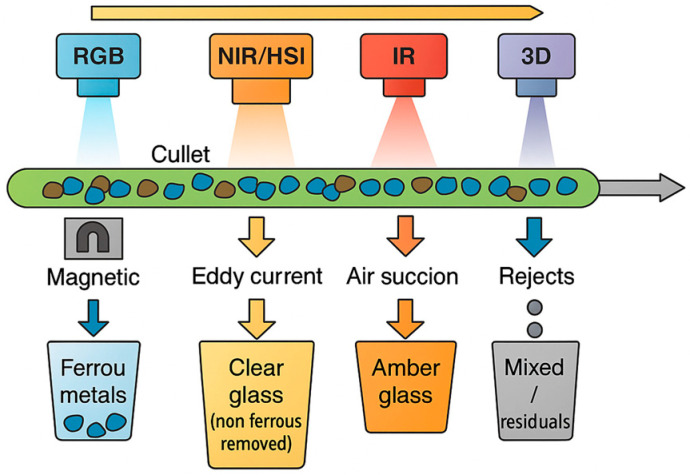
Schematic arrangement of sensors along a glass waste conveyor line.

**Figure 5 sensors-25-06634-f005:**
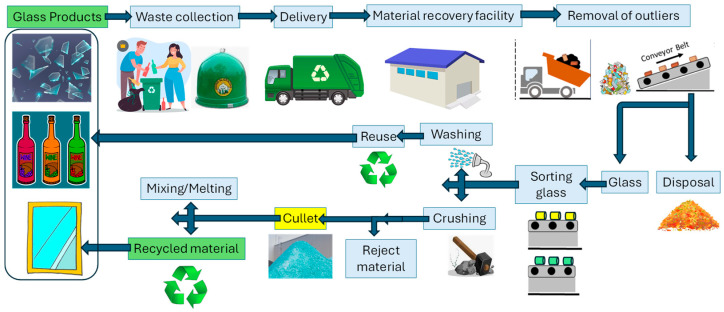
Stages in glass waste processing.

**Figure 6 sensors-25-06634-f006:**
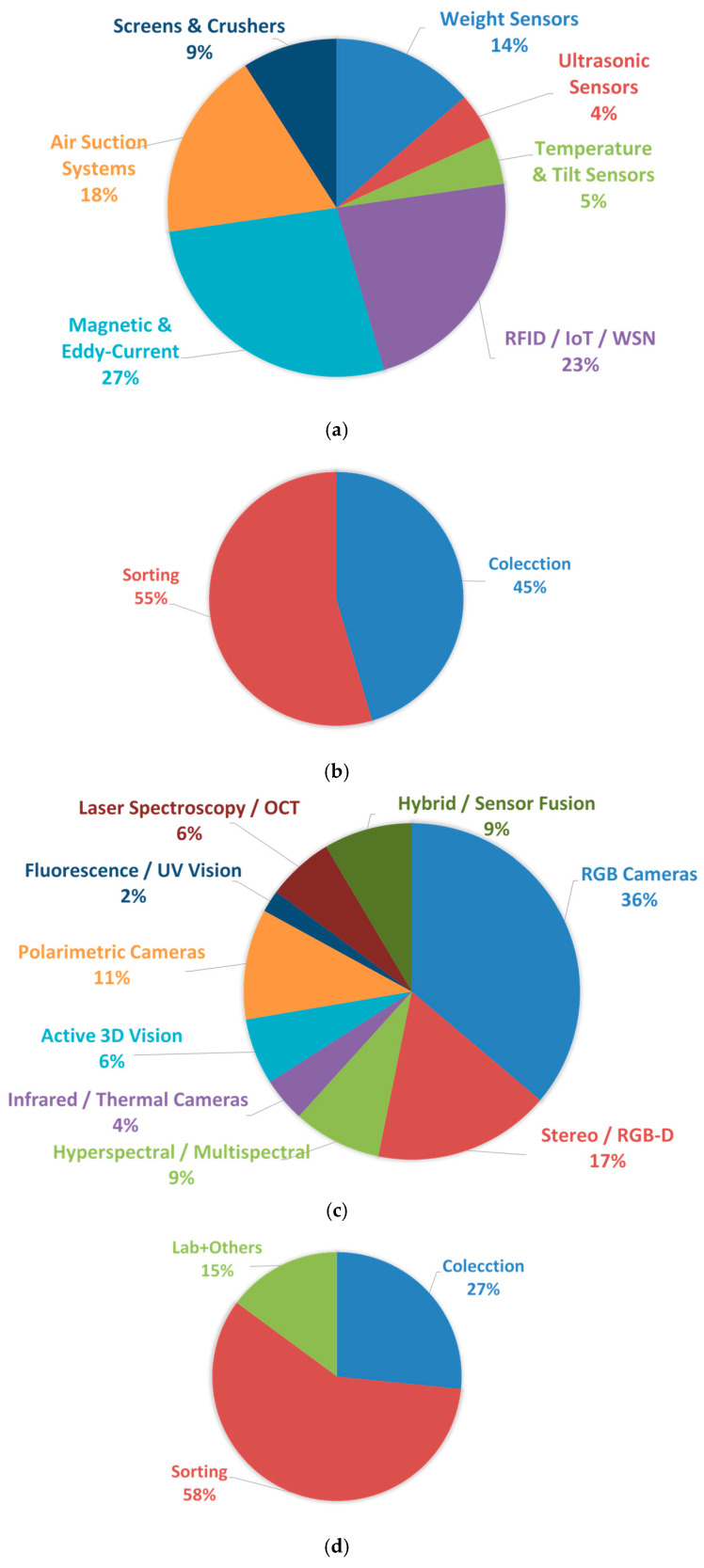
(**a**,**b**) Relative frequencies of use and percentages of plant location types for non-vision technologies; (**c**,**d**) relative frequencies of use and percentage distribution of plant location types for vision-based technologies.

**Table 1 sensors-25-06634-t001:** Search string formulation for the review of scientific papers.

Item	Reason	Target	Search String
1	Type of waste	Glass waste	Waste AND Glass
2	Interaction with waste	Identification	Glass AND (recognition OR collection OR recycling)
3	General device interaction (excluding CV)	Sensors	Glass AND (robotics OR ultrasonics OR communication networks OR magnetics OR air suction)
4	Computer vision devices	Vision-based sensors	Glass AND (camera OR scanner OR 3D system OR hyperspectral)

**Table 2 sensors-25-06634-t002:** Representative studies that use different equipment and technologies for glass waste management (excluding computer vision).

Technology	Ref.	Publication	Enviro.	Glass and…	Stage	Objective
Trucks	Yuan et al. [38] (A1)	Journal	Public spaces	Plastic, metals	Collection	Optimize automated collection using robotic arms
Robots	Ogawa et al. [39] (A1)	Web/Industry	Public spaces	Mixed waste	Collection	Automate collection with robots and IoT sensors
Weight sensors	Brouwer et al. [20] (A1)	Journal	Commercial areas	General urban waste	Collection	Logistic control and waste weighing
Ultrasonic sensors	Yerraboina et al. [21] (A1)	Journal	Public spaces	Not specified	Collection	Measure container fill level
Thermal probe	Rovetta et al. [22] (A1)	Journal	Public spaces	Not specified	Collection	Alert for thermal conditions or tipping
Magnetic separator	Bellopede et al. [4] (A1)	Journal	Industrial areas	Metals, ceramics	Separation of other materials	Remove metallic contaminants from glass
Magnetic pulleys	Eriez [51] (B3)	Web	Industrial areas	Metals	Separation of other materials	Fine separation of metallic particles
Eddy current separators	Steinert [24] (B3)	Web	Industrial areas	Aluminium, brass	Separation of other materials	Remove non-ferrous metals like aluminium
Air suction	ZZS [25] (B3)	Web/Industry	Industrial areas	Paper, plastics	Separation of other materials	Remove light contaminants such as paper or plastic
Sieves	Efremenkov [26] (A1)	Journal	Industrial areas	Plastics, metals, organics	Classification	Separate cullet into specific particle size fractions
Communication networks	Hannan et al. [23] (A1)	Journal	Public spaces	Urban solid waste	Collection	Optimize routes and collection using RFID, GPS, GSM

**Table 3 sensors-25-06634-t003:** Representative studies that use computer vision-based equipment and technologies for glass waste management.

Technology	Ref.	Publication	Environ.	Glass and…	Stage	Objective
RGB cameras	Cheng et al. [72] (A1)	Journal	Industrial areas	Glass, paper, cardboard, plastic, metal	Classification	Automated classification of glass bottles within mixed waste
Spectral/Hyperspectral c.	Bonifazi et al. [31] (A1)	Journal	Industrial areas	Ceramics, glass	Classification	Spectral recognition of glass and ceramic materials
Infrared (thermal) cameras	Huo et al. [32] (A1)	Journal	Industrial areas	Mixed waste	Recognition	Reliable glass segmentation using thermal and RGB imaging
RGB stereo cameras	Wu et al. [29] (A1)	Journal	Households	Mixed recyclable waste	Classification	Three-dimensional characterization of recyclable objects
3D active vision systems	[33] (B3)	Web/Industry	Industrial areas	Dark glass, contaminants	Classification	Advanced glass identification with structured light and HSI sensors
Polarimetry	Taglione et al. [34] (A1)	Journal	Public spaces/Robotics	Glass, plastics, metals	Recognition	Distinguish transparent objects such as glass by polarization signature
Fluorescence/UV vision	Reinhold et al. [35] (A2)	Patent	Industrial areas	Glass with additives, lead	Classification	Identification of special glass using UV fluorescence
Optical/laser tomography	Pontes et al. [36] (A1)	Journal	Industrial areas	Technical glass	Recognition	Characterization of type and granulometry using LIBS
Hybrid technologies	Casao et al. [37] (A1)	Journal	Industrial areas	Glass, plastic, paper	Classification	Improved classification through fusion of RGB and spectral sensors

**Table 4 sensors-25-06634-t004:** Summary of publicly available image datasets containing glass-related objects, organized into six categories: Packaging and Containers (T1), Tableware and Kitchenware (T2), Household and Decorative Items (T3), Construction and Architecture (T4), Technical and Industrial Glass (T5), and Automotive and Transport (T6). All references are B3: Vendor-based source (B) and Web page (3). The highest percentage ranges are highlighted in bold.

		Estimated Range by Category (%)							
Ref. B3	Aprox. Size	T1	T2	T3	T4	T5	T6	Company/Authors	Most Frequent Cases
[111]	12,500	**35–40**	4–7	2–4	25–30	20–25	12–15	DataCluster Labs	Glass bottles, jars, ampoules, vials. Windows, façades, doors, shower screens
[112]	297,500	**25–30**	15–20	10–15	20–25	15–20	10–15	Roboflow Universe	Bottles, jars, vials, flasks, ampoules (e.g., beverage bottles, cosmetic containers. Windows, doors, façades, shower screens, glass blocks
[113]	4150	35–40	0	0	0	**55–60**	1–5	MVTec Software GmbH	Flat glass sheets with surface defects (scratches, bubbles, inclusions). Inspection of glass bottles for cracks, scratches, contamination
[114]	9500	25–30	10–12	8–10	**35–40**	5–7	3–5	Enze Xie et al.	Windows, doors, partitions, glass walls. Transparent bottles, jars, cosmetic containers, drink packaging
[115]	2925	**25–30**	10–15	10–15	20–25	10–15	5–10	Datacluster-labs	Transparent bottles, jars, drink containers. Window panes, glass walls, partitions
[116]	1050	10–15	0	0	0	**80–85**	0	Sagieppel. MIT	Laboratory glassware (beakers, flasks, test tubes, cylinders) used in chemistry experiments
[117]	1275	**70–75**	5–10	3–7	5–10	7–12	10–14	TrashIVL Dataset	Glass bottles used as recyclable waste
[118]	1935	**40–45**	5–7	3–5	5–10	7–10	5–8	TashBox Dataset	Glass bottles, jars, containers in waste

## Data Availability

No new data were created or analyzed in this study. Data sharing is not applicable to this article.

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
