# Peer review of "Computer Vision for Glass Waste: Technologies and Sensors"

_sensors, 2025, doi:10.3390/s25216634_

Round 1

Reviewer 1 Report

Comments and Suggestions for Authors

The manuscript "Computer Vision for Glass Waste: Technologies and Sensors" is comprehensive and addresses a clear gap in the literature. Its main strengths lies in the coverage of glass-specific waste, and inclusion of both academic and industrial perspectives. There are some few comments to help improve the manuscript.

  1. The introduction provides a strong rationale for focusing on glass waste as a distinct stream within the waste recycling field of research, however, it is overly broad at first, spending much space on general solid waste management rather than quickly narrowing down to glass and computer vision. Focus should shift early to narrowing down to glass waste.
  2. The manuscript says Scopus, Web of Science plus google scholar were used, but it’s not clear how inclusion/exclusion was finalized. The inclusion and exclusion criteria are not fully detailed, and it is unclear whether a systematic review framework such as PRISMA was used.

Author Response

Reviewer #1

The manuscript "Computer Vision for Glass Waste: Technologies and Sensors" is comprehensive and addresses a clear gap in the literature. Its main strengths lies in the coverage of glass-specific waste, and inclusion of both academic and industrial perspectives. There are a few comments to help improve the manuscript.

We thank the reviewer for the constructive remarks and recommendations. First of all, we would like to inform the reviewer that this new version of the paper already incorporates a previous round of revisions based on the Editor’s suggestions. Therefore, this submission represents the second revised version of the manuscript.

The introduction provides a strong rationale for focusing on glass waste as a distinct stream within the waste recycling field of research, however, it is overly broad at first, spending much space on general solid waste management rather than quickly narrowing down to glass and computer vision. Focus should shift early to narrowing down to glass waste.

Response:

Thank you for this valuable suggestion. We acknowledge that the previous structure resulted in an excessively long preamble with an extensive theoretical background in Sections 1 and 2. Therefore, these two sections have been merged and condensed into a new Section 1, titled Introduction: Glass Waste and Recycling. The topics of glass and computer vision are addressed directly from Sections 4 to 8. Section 3 has also been revised to clearly highlight the objectives and contributions of the paper.

The manuscript says Scopus, Web of Science plus google scholar were used, but it’s not clear how inclusion/exclusion was finalized. The inclusion and exclusion criteria are not fully detailed, and it is unclear whether a systematic review framework such as PRISMA was used.

Response

Yes. In the context of the databases Scopus, Web of Science, and Google Scholar, the inclusion and exclusion criteria were based on the Preferred Reporting Items for Systematic Reviews and Meta-Analyses (PRISMA) framework. Section 3, Methodological Aspects of the Review, has been partially rewritten to include these criteria.

Reviewer 2 Report

Comments and Suggestions for Authors

The review “Computer Vision for Glass Waste: Technologies and Sensors” is well-organized, well-written and easy to understand. The manuscript is clear, and relevant for the field of waste management. The review contains eight well-developed sections: 1) Global Overview of Solid Waste, 2) Representative Figures on Glass Waste Generation and Recycling, 3) Objectives, contributions and structure of the paper, 4) Methodological Aspects of the Review, 5) Equipment and Technologies for Glass Waste Management (Excluding Computer Vision), 6) Computer Vision-Based Technologies for Glass Waste Management, 7) Brief Remarks on Algorithms and Public Image Datasets for Glass Waste, 8) Discussion. Conclusion is missing. Strong suggestion is to extract conclusion from Discussion chapter.

The main question by the review is addressed the technologies specifically involved in the collection and separation stages of glass waste that are related to laboratory developments or experiments on standard datasets, but also includes projects, patents, and real world implementations.

The authors do a good job of synthesizing the literature. The review is clear, comprehensive and relevant in glass collection and recycling, highlighting the latest applied technologies, and focusing on computer vision–based approaches. The cited references are mostly recent publications (73% within the last 5 years) and do not include any self-citations. Although several reviews on solid waste and recycling have been published, generally covering all types of materials, there are hardly any review articles focusing specifically on glass. Gap in knowledge is particularly noted, with no article in the literature containing comprehensive review of the state of the art in glass collection and recycling with an emphasis on computer vision–based approaches.

The topic is original and its relevance is reinforced by the fact that glass constitutes approximately 10% of the world’s waste material. The novelty of this review stems from beforementioned gap.

Compared to other published material this review contributes to the subject area not only by showcasing the most effective computer vision technologies and devices currently employed in glass collection and sorting processes but also by uncovering their limitations, gaps, and shortcomings.

The methodology is clearly explained and an initial survey of the literature is conduced, covering topics related to glass waste and the technologies applied across the different stages of its treatment. Searches were carried out in the areas of waste acquisition or collection, classification and recognition, and glass recycling. These results were subsequently refined in order to establish more specific objectives, focusing on sensors and techniques associated with robotics and automation. Finally, there were total of 258 articles concerning glass and after screening the titles and abstracts, non-relevant works were excluded, resulting in 81 studies directly related to waste management. Studies have shown that automated computer vision techniques significantly reduce processing times and enhance efficiency compared to manual methods, particularly in the acquisition and classification of glass waste

The discussion is well supported by the comprehensive review and arguments of computer vision technologies, which includes a concise introduction to software technologies, highlighting their potential applications, advantages, and limitations. This article has demonstrated how current computer vision-based technologies can automate complex sorting tasks, reduce operational costs, and improve the quality of recovered glass materials, thereby contributing to the circular economy.

Diagrams and chart are easy to interpret and understand. Three figures are comprehensive and adequate. Three tables interpreted data consistently and appropriately.

Author Response

We thank the reviewer for the constructive comments. We understand that no further modifications to the paper are required.

The review “Computer Vision for Glass Waste: Technologies and Sensors” is well-organized, well-written and easy to understand. The manuscript is clear, and relevant for the field of waste management. The review contains eight well-developed sections: 1) Global Overview of Solid Waste, 2) Representative Figures on Glass Waste Generation and Recycling, 3) Objectives, contributions and structure of the paper, 4) Methodological Aspects of the Review, 5) Equipment and Technologies for Glass Waste Management (Excluding Computer Vision), 6) Computer Vision-Based Technologies for Glass Waste Management, 7) Brief Remarks on Algorithms and Public Image Datasets for Glass Waste, 8) Discussion. Conclusion is missing. Strong suggestion is to extract conclusion from Discussion chapter.

The main question by the review is addressed the technologies specifically involved in the collection and separation stages of glass waste that are related to laboratory developments or experiments on standard datasets, but also includes projects, patents, and real world implementations.

The authors do a good job of synthesizing the literature. The review is clear, comprehensive and relevant in glass collection and recycling, highlighting the latest applied technologies, and focusing on computer vision–based approaches. The cited references are mostly recent publications (73% within the last 5 years) and do not include any self-citations. Although several reviews on solid waste and recycling have been published, generally covering all types of materials, there are hardly any review articles focusing specifically on glass. Gap in knowledge is particularly noted, with no article in the literature containing comprehensive review of the state of the art in glass collection and recycling with an emphasis on computer vision–based approaches.

The topic is original and its relevance is reinforced by the fact that glass constitutes approximately 10% of the world’s waste material. The novelty of this review stems from beforementioned gap.

Compared to other published material this review contributes to the subject area not only by showcasing the most effective computer vision technologies and devices currently employed in glass collection and sorting processes but also by uncovering their limitations, gaps, and shortcomings.

The methodology is clearly explained and an initial survey of the literature is conduced, covering topics related to glass waste and the technologies applied across the different stages of its treatment. Searches were carried out in the areas of waste acquisition or collection, classification and recognition, and glass recycling. These results were subsequently refined in order to establish more specific objectives, focusing on sensors and techniques associated with robotics and automation. Finally, there were total of 258 articles concerning glass and after screening the titles and abstracts, non-relevant works were excluded, resulting in 81 studies directly related to waste management. Studies have shown that automated computer vision techniques significantly reduce processing times and enhance efficiency compared to manual methods, particularly in the acquisition and classification of glass waste

The discussion is well supported by the comprehensive review and arguments of computer vision technologies, which includes a concise introduction to software technologies, highlighting their potential applications, advantages, and limitations. This article has demonstrated how current computer vision-based technologies can automate complex sorting tasks, reduce operational costs, and improve the quality of recovered glass materials, thereby contributing to the circular economy.

Diagrams and chart are easy to interpret and understand. Three figures are comprehensive and adequate. Three tables interpreted data consistently and appropriately.

Reviewer 3 Report

Comments and Suggestions for Authors

The article is a literature review on technologies related to glass waste. It addresses technologies for collecting, separating, and classifying glass waste, as well as datasets, algorithms, and computer vision-based technologies for analyzing glass waste.

*While the article is generally well written, there are some details that could be improved to achieve a more formal style. For example, the sentence “These works often highlight the role of current technologies in mitigating recycling …” could be reformulated as “These studies often emphasize the role of existing technologies in addressing recycling …”.

Therefore, I recommend revising the writing to enhance its formality.

*The first three sections could be combined as a single Introduction, since the current structure creates a very long preamble with extensive theoretical background. A research article should be written with a stronger emphasis on the contribution. I suggest reducing the content of Section 1 and merging Sections 1, 2, and 3 into a single section with the Introduction.

*The captions of Figures 1 and 2 are somewhat small. I recommend redesigning these figures to make the captions easier to read.

*Although Sections 5 and 6 provide a good description of technologies and sensors, the paper would benefit from an additional section focused on statistical analysis. This could include a diagram summarizing the frequency of use of the different technologies presented and indicating the stage of the glass waste processing cycle where they have the most impact. Such an addition would strengthen both the Discussion and Conclusions, as they could be supported by quantitative results.

*Although 100 references represent a substantial number of works identified using the methodology described in Section 4, I believe that a comprehensive review of the state of the art and current technologies requires a significantly larger set of references. I therefore suggest conducting a more exhaustive search to include at least 20 additional references that reinforce the ideas and statistics presented.

Comments on the Quality of English Language

I recommend revising the writing to enhance its formality.

Author Response

Reviewer #3

The article is a literature review on technologies related to glass waste. It addresses technologies for collecting, separating, and classifying glass waste, as well as datasets, algorithms, and computer vision-based technologies for analyzing glass waste.

We thank the reviewer for the constructive remarks and suggestions. First of all, we would like to inform the reviewer that this new version of the paper already incorporates a previous round of revisions based on the Editor’s suggestions. Therefore, this submission represents the second revised version of the manuscript.

*While the article is generally well written, there are some details that could be improved to achieve a more formal style. For example, the sentence “These works often highlight the role of current technologies in mitigating recycling …” could be reformulated as “These studies often emphasize the role of existing technologies in addressing recycling …”.

Therefore, I recommend revising the writing to enhance its formality.

Response:

Thank you for this helpful suggestion. We have revised and improved the wording accordingly.

*The first three sections could be combined as a single Introduction, since the current structure creates a very long preamble with extensive theoretical background. A research article should be written with a stronger emphasis on the contribution. I suggest reducing the content of Section 1 and merging Sections 1, 2, and 3 into a single section with the Introduction.

Response.

Thank you for this valuable suggestion. We acknowledge that the previous structure resulted in an excessively long preamble with an extensive theoretical background in Sections 1 and 2. Therefore, these sections have been merged and condensed into a new Section 1, titled Introduction: Glass Waste and Recycling. The former Section 3 (now Section 2) has also been revised to clearly highlight the objectives and contributions of the paper.

*The captions of Figures 1 and 2 are somewhat small. I recommend redesigning these figures to make the captions easier to read.

Response.

The captions of Figures 1 and 2 have been improved and are now easier to read.

*Although Sections 5 and 6 provide a good description of technologies and sensors, the paper would benefit from an additional section focused on statistical analysis. This could include a diagram summarizing the frequency of use of the different technologies presented and indicating the stage of the glass waste processing cycle where they have the most impact. Such an addition would strengthen both the Discussion and Conclusions, as they could be supported by quantitative results.

Response

We agree with the reviewer that this would be an interesting aspect to explore. However, conducting a worldwide study on the current use of each technology would be a complex task that cannot be accomplished within the ten-day period indicated by the editor. Nevertheless, we have begun an extensive review and found no peer-reviewed publications providing global percentages or validated data on the frequency of use of these technologies. For the moment, we have not been able to implement this valuable suggestion, but we believe it could be thoroughly addressed in a future study.
We would like to highlight that a substantial number of changes have been introduced in the paper based on the reviewers’ comments, including new requested information and the reduction of other parts. New sections, references, figures, studies, and tables have also been added. We therefore believe that the manuscript has been significantly improved.

*Although 100 references represent a substantial number of works identified using the methodology described in Section 4, I believe that a comprehensive review of the state of the art and current technologies requires a significantly larger set of references. I therefore suggest conducting a more exhaustive search to include at least 20 additional references that reinforce the ideas and statistics presented.

Response

We agree that a more exhaustive search would reinforce the strength of the paper. Following the reviewer’s recommendations, we have incorporated 25 additional references into the article. We believe that the inclusion of these references has significantly strengthened the overall documentation and scholarly foundation of the paper.

Reviewer 4 Report

Comments and Suggestions for Authors

The paper fills a real gap by focusing specifically on glass in waste-management pipelines and surveys both non-vision equipment (Section 4; Figure 3; Table 1) and computer-vision/sensing modalities (Section 5; Table 2), with a short bridge to algorithms and datasets (Section 6). Some quantitative claims and scope boundaries would benefit from clarification before acceptance, including the following:

1) Please explain more explicitly what is novel in this review beyond existing broad waste-sorting surveys and the recent glass-focused review cited as [15]. In Section 3 you promise that Part 2 will cover methods/algorithms; please delineate the present article’s unique contributions (e.g., a taxonomy of imaging sensors by task/setting, industrial performance synthesis) and add a short “what this review adds” paragraph at the end of Section 1 or 3.  

2) Please reconcile the different shares and rates reported for glass. Specify scope, geography, and year for each value, and align Figure 2’s bars with named sources and years in the caption.

3) Consider presenting a single table summarizing % glass and recycling rates by region with sources.

4) Please substantiate performance statements with definitions and conditions e.g. Section 5 reports SEEGLASS achieving 99% purity and 85% recovery, and Section 7.1 mentions “accuracy above 95%” in real-world environments. Clarify datasets, test conditions (belt speed, lighting, mix), metric definitions (purity, recovery, accuracy, precision/recall), confidence intervals, and whether results are vendor, project-deliverable, or peer-reviewed. Where claims stem from web/patent/project pages, label them as such in text. 

5) Please harmonize naming and acronyms throughout t (e.g., “RGB cameras” vs “Cameras RGB” etc). 

6) Consider consider adding a one-page taxonomy figure that maps sensing modality → physical principle - typical target (e.g., colour sorting, ceramic/metal contamination, glass vs transparent plastic)-  plant location (collection vs MRF line) with example systems from Table 1/2. A small schematic showing sensor placement around a conveyor would greatly help readers connect Sections 4-5 with Figure 3. 

7) Please consider expanding Section 6 to discuss industrial datasets for cullet streams, the minimum viable benchmark for glass vs transparent plastics (class definitions, illumination protocols, belt speeds, occlusion/overlap), and data-sharing or synthetic-data strategies. A short table listing public datasets that actually include glass in realistic settings (beyond clean-background TrashNet/TACO) and what is missing would make your “future directions” section (7.2) more actionable.

Author Response

Reviewer #4

The paper fills a real gap by focusing specifically on glass in waste-management pipelines and surveys both non-vision equipment (Section 4; Figure 3; Table 1) and computer-vision/sensing modalities (Section 5; Table 2), with a short bridge to algorithms and datasets (Section 6). Some quantitative claims and scope boundaries would benefit from clarification before acceptance, including the following:

We thank the reviewer for the constructive remarks and suggestions. First of all, we would like to inform the reviewer that this new version of the paper already incorporates a previous round of revisions based on the Editor’s suggestions. Therefore, this submission represents the second revised version of the manuscript.

1) Please explain more explicitly what is novel in this review beyond existing broad waste-sorting surveys and the recent glass-focused review cited as [15]. In Section 3 you promise that Part 2 will cover methods/algorithms; please delineate the present article’s unique contributions (e.g., a taxonomy of imaging sensors by task/setting, industrial performance synthesis) and add a short “what this review adds” paragraph at the end of Section 1 or 3. 

Response

Thank you for this valuable suggestion. The former Section 3 (now Section 2) has been revised to emphasize the contributions and novelty of this paper. A critical commentary on the work by Baek et al. has been added, and the scope of the present review has been clearly defined by excluding the planned Part 2, which will be addressed in a future publication. These revisions can be found in paragraphs 2 and 3 of the new Section 2.

2) Please reconcile the different shares and rates reported for glass. Specify scope, geography, and year for each value, and align Figure 2’s bars with named sources and years in the caption. 3) Consider presenting a single table summarizing % glass and recycling rates by region with sources.

Response

The former Section 2 (now Section 1.2) has been revised and updated with new data from articles published mainly within the last two years. Likewise, a new Figure 2 has been created following the reviewer’s suggestions.
We would be open to including additional collateral data; however, we consider that the former Sections 1 and 2 already provide sufficient quantitative information, and that further figures are not essential to the core of the paper. In fact, two reviewers emphasized that the first three sections could be combined and possibly reduced, as they felt that the previous structure resulted in a very long preamble with extensive theoretical background. Following these suggestions, the former Sections 1 and 2 (now Sections 1.1 and 1.2) have been condensed and merged into Section 1: Introduction – Glass Waste and Recycling.

4) Please substantiate performance statements with definitions and conditions e.g. Section 5 reports SEEGLASS achieving 99% purity and 85% recovery, and Section 7.1 mentions “accuracy above 95%” in real-world environments. Clarify datasets, test conditions (belt speed, lighting, mix), metric definitions (purity, recovery, accuracy, precision/recall), confidence intervals, and whether results are vendor, project-deliverable, or peer-reviewed. Where claims stem from web/patent/project pages, label them as such in text.

Response

We have carefully revised the manuscript to address the reviewer’s suggestions. In this regard, we have clarified the meaning of the performance metrics used in the text (such as purity, recovery, accuracy, precision, and recall), added relevant references and explicitly indicated the origin of each reported result (e.g., vendor-based, project deliverable, or peer-reviewed source), as well as whether it comes from a scientific journal or conference, patent, or project/web page.
To provide clearer information to the reader, in Sections 4 to 8 we have added a set of acronyms as subscripts to each reference to indicate its origin — Peer-reviewed source / Book (A), Vendor-based source (B), Project deliverable (C) — and whether the information comes from a scientific journal or conference (1), a patent (2), or a web page (3). Accordingly, each reference appears as follows [XX]A1

5) Please harmonize naming and acronyms throughout t (e.g., “RGB cameras” vs “Cameras RGB” etc).

Response:

We appreciate this helpful comment. The manuscript has been thoroughly reviewed to harmonize the use of names and acronyms throughout the text.

6) Consider consider adding a one-page taxonomy figure that maps sensing modality → physical principle - typical target (e.g., colour sorting, ceramic/metal contamination, glass vs transparent plastic)-  plant location (collection vs MRF line) with example systems from Table 1/2. A small schematic showing sensor placement around a conveyor would greatly help readers connect Sections 4-5 with Figure 3.

Response

Thank you for this excellent suggestion. A new Section 4, entitled “Overview of Equipment and Technologies,” has been added to introduce the one-page taxonomy that summarizes the main sensing modalities and their relationships (see Figure 3). In this section, we have also included the small schematic illustrating the sensor placement around a conveyor, as suggested by the reviewer (see Figure 4). We believe that this new section effectively addresses the gap identified by the reviewer and improves the overall structure and readability of the paper.

7) Please consider expanding Section 6 to discuss industrial datasets for cullet streams, the minimum viable benchmark for glass vs transparent plastics (class definitions, illumination protocols, belt speeds, occlusion/overlap), and data-sharing or synthetic-data strategies. A short table listing public datasets that actually include glass in realistic settings (beyond clean-background TrashNet/TACO) and what is missing would make your “future directions” section (7.2) more actionable.

Thank you for this very interesting comment. We have carefully reviewed the former Section 6 (now Section 7) with the intention of briefly expanding some aspects suggested by the reviewer. In order to complement this section, Table 4 in Section 7.2 presents publicly available datasets that include glass in realistic settings—beyond clean-background conditions—and that could be applied in industrial environments. These datasets are organized into six categories: Packaging and Containers (T1), Tableware and Kitchenware (T2), Household and Decorative Items (T3), Construction and Architecture (T4), Technical and Industrial Glass (T5), and Automotive and Transport (T6). For each dataset, we report the approximate dataset size, the estimated percentage range of images corresponding to each category, the institution or authors responsible for the dataset, and a brief description of the most common objects included in the image collection.

Please note that, as stated at the beginning of this section, the focus of this review is exclusively on imaging sensor technologies for glass waste treatment. In fact, the section is entitled “Brief Remarks on Algorithms and Public Image Datasets for Glass Waste.” Therefore, the additional topics and analyses suggested by the reviewer will be addressed in a forthcoming review specifically dedicated to datasets and algorithms, where a more exhaustive study can be conducted.
Regarding the discussion on the minimum reference requirements for differentiating glass from transparent plastics (class definitions, illumination protocols, conveyor speeds, occlusion/overlap conditions, etc.), we consider this a specific topic that will certainly be examined in the forthcoming review.

Round 2

Reviewer 3 Report

Comments and Suggestions for Authors

The authors have addressed almost all of the previous comments. Only one observation remains:

I still insist that the paper would benefit from an additional section focused on statistical analysis. This could include a diagram summarizing the frequency of use of the different technologies presented, indicating the stage of the glass waste processing cycle where each has the greatest impact. Such an addition would strengthen both the Discussion and Conclusions, as these sections could then be supported by quantitative evidence.

However, this statistical analysis can be based on the findings from the state-of-the-art review already conducted; it is not necessary to search a literature review paper or a worldwide analysis. I believe that with 125 references, a meaningful statistical sampling can be achieved.

Author Response

Response to Reviewer #3

The authors have addressed almost all of the previous comments. Only one observation remains:

I still insist that the paper would benefit from an additional section focused on statistical analysis. This could include a diagram summarizing the frequency of use of the different technologies presented, indicating the stage of the glass waste processing cycle where each has the greatest impact. Such an addition would strengthen both the Discussion and Conclusions, as these sections could then be supported by quantitative evidence.

However, this statistical analysis can be based on the findings from the state-of-the-art review already conducted; it is not necessary to search a literature review paper or a worldwide analysis. I believe that with 125 references, a meaningful statistical sampling can be achieved.

Response:

Thank you once again for your invaluable contribution to improving the quality of our article. We have added new Section 8, Statistical Analysis of Technology Use, and included a summary diagram (new Figure 6). Based on the literature reviewed in this paper, this figure — consistent with Figure 3 — shows the relative frequency of use of each technology and the percentage distribution of plant location types for non-vision and vision-based technologies. In addition, we have included comments discussing the results of this statistical analysis.